# Organic–Inorganic Nanohybrid Electrochemical Sensors from Multi-Walled Carbon Nanotubes Decorated with Zinc Oxide Nanoparticles and In-Situ Wrapped with Poly(2-methacryloyloxyethyl ferrocenecarboxylate) for Detection of the Content of Food Additives

**DOI:** 10.3390/nano9101388

**Published:** 2019-09-27

**Authors:** Jing-Wen Xu, Zhuo-Miao Cui, Zhan-Qing Liu, Feng Xu, Ya-Shao Chen, Yan-Ling Luo

**Affiliations:** 1Key Laboratory of Macromolecular Science of Shaanxi Province, School of Chemistry and Chemical Engineering, Shaanxi Normal University, Xi’an 710062, Chinayschen@snnu.edu.cn (Y.-S.C.); 2School of Food & Biological Engineering, Shaanxi University of Science and Technology, Xi’an 710021, China; 3Shaanxi Province Engineering Research Center of Coal Conversion Alcohol, College of Chemistry and materials, Weinan Normal University, Weinan 710114, China

**Keywords:** nanomaterials, carbon nanotubes, electrochemical properties, ferrocene derivatives, sensors

## Abstract

An electrochemical sensor for detection of the content of aspartame was developed by modifying a glassy carbon electrode (GCE) with multi-walled carbon nanotubes decorated with zinc oxide nanoparticles and in-situ wrapped with poly(2-methacryloyloxyethyl ferrocenecarboxylate) (MWCNTs@ZnO/PMAEFc). MWCNTs@ZnO/PMAEFc nanohybrids were prepared through reaction of zinc acetate dihydrate with LiOH·H_2_O, followed by reversible addition-fragmentation chain transfer polymerization of 2-methacryloyloxyethyl ferrocenecarboxylate, and were characterized by Fourier transform infrared spectroscopy (FTIR), thermogravimetric analysis (TGA), Raman, X-ray diffraction (XRD), X-ray photoelectron spectroscopy (XPS), atomic force microscope (AFM), scanning electron microscope (SEM), and transmission electron microscope (TEM) techniques. The electrochemical properties of the prepared nanohybrids with various composition ratios were examined by cyclic voltammetry (CV), and the trace additives in food and/or beverage was detected by using differential pulse voltammetry (DPV). The experimental results indicated that the prepared nanohybrids for fabrication of electrochemical modified electrodes possess active electroresponse, marked redox current, and good electrochemical reversibility, which could be mediated by changing the system formulations. The nanohybrid modified electrode sensors had a good peak current linear dependence on the analyte concentration with a wide detection range and a limit of detection as low as about 1.35 × 10^−9^ mol L^−1^, and the amount of aspartame was measured to be 35.36 and 40.20 µM in Coke zero, and Sprite zero, respectively. Therefore, the developed nanohybrids can potentially be used to fabricate novel electrochemical sensors for applications in the detection of beverage and food safety.

## 1. Introduction

Aspartame is a dipeptide artificial sweetener, and extensively used in the manufacturing of many sugar-free, low calorie, and dietary products [1,2]. It has been reported that aspartame can be absorbed by the human body completely, and does not cause any harm to human body, and thus is considered to be a safe and reliable additive [3]. Even if the methanol disengages due to hydrolysis, the extremely small intake of methanol does not harm the human body because the amount of aspartame is extremely low [4,5,6]. Nevertheless, its safety has been questioned after all in that it is suspected if aspartic acid in aspartame would cause brain damage, endocrine disorders, or tumors. In particular, aspartame as a main ingredient of many children’s food possesses greater dangerousness for children. For pregnant women, adverse effects on the health and fetus may be caused when they have eaten food containing aspartame or take drugs containing the ingredient. The acceptable daily intake (ADI) of aspartame is currently 50 mg (kg body weight)^−1^ (kg b. wt.)**^-1^** in USA and 40 mg (kg b. wt.)^−1^ in Europe. Most extensively adopted detection methods for aspartame have capillary electrophoreses [7,8] and high-performance liquid chromatography (HPLC) techniques [9]. These techniques have satisfying selectivity and limit of detection (LOD); the drawback is however time-consuming and needs expensive instruments and pre-treatment steps in comparison with the electrochemical analyses that have the advantage of quick response, high sensitivity, and simple operation [10]. Electronic tongues and electronic noses have received increasing attention as these techniques gain more and more applicability in pharmaceutical industry including sweeteners analysis [11,12]. Nevertheless, their sensitivity, specificity and response range are yet to be improved. This offers us a strong impetus in engineering and developing new sensitive and selective aspartame detection techniques for food safety detection and consumer protection.

Up to now, the electrode modification remains an effective method to improve the sensitivity. Since carbon nanotubes (CNTs) exhibit the activity of edge-plane-like graphite sites at the CNTs ends, they can promote the electron transfer in electroanalytical applications [13], and the electrochemical (bio)sensors based on MWCNTs exhibit high sensitivity and low LOD for food safety detection or quality control [14,15]. On the other hand, MWCNTs can be modified and functionalized by anchoring some nanoparticles (NPs)—such as zinc oxide (ZnO), gold, and silver nanoparticles—reduced graphene on their surface or their external walls [15,16,17,18]. These nanomaterials are widely used to fabricate electrochemical sensors due to their excellent electrochemical properties. They can enlarge the electrode superficial area, further increase the electron transfer rate, and ultimately improve the sensitivity of the electrochemical (bio)sensors [15,16,17,18,19,20]. It has been proved that the electrochemical (bio)sensors, particularly sensors based on screen printed technology, are highly sensitive, easily designable and cost-effective, and readily-miniaturized [12,21,22,23], but some issues remain to be improved. It is still an important and challenged work for us how to achieve a better electrochemical response and repeatedly stability, minimum LOD, and wider detection range for development of electrochemical sensors, whereas selecting and optimizing sensing materials with unique comprehensive performances may be one of the potentially effective routes. Zinc oxide (ZnO) is a semiconducting material, which possesses a wide and direct band gap of about 3.37 eV at 300 K, a large excitation binding energy of 60 MeV, high mechanical and thermal stabilities, and radiation hardness [24]. Particularly, the nano-scaled ZnO particles have unique properties such as subsize, large superficial area, high selectivity and sensitivity, semiconductivity, biocompatibility, and chemical durability [25,26]; ZnO nanoparticles (ZnO NPs) can promote the electron transfer process at the interface of electrodes and solutions [27]. These benefits mean ZnO NPs are widely used for the establishment and/or fabrication of electrochemical sensors with improved analytical performances. It was reported that ZnO NPs could be hybridized with MWCNTs to fabricate electrochemical sensors for improving the electrochemical signal, lowering LOD, and determining aspartame in food and beverage samples [28]. Ferrocene and its derivatives have attracted significant interest in the area of electrochemical (bio)sensing, semiconducting materials, electrocatalysis etc., due to their excellent electrical activities, fast electron transfer properties, and good electrochemical response [29,30,31,32]. However, the use of any single component above mentioned does not achieve the expected effect in our study. The designability and anisotropy of the properties of composite materials, especially their combined effect—including unique additivity, productivity, modularity, and seepage behavior [33]—inspire us to effectively combine these components to construct a novel hybrid with the enhanced electron transfer rate, the improved electrochemical properties, and the minimized LOD value. In fact, a voltammetric sensor based on a ZnO NPs/ferrocene derivative modified carbon paste electrode was reported for detection of captopril in drug samples, which proved high sensitivity, good selectivity, and rapidity [34].

Inspired by the above description, our aim is to prepare MWCNTs scattered with ZnO (MWCNTs@ZnO) nanocomposites, and then to wrap poly(2-methacryloyloxyethyl ferrocenecarboxylate) (PMAEFc) around MWCNTs@ZnO to achieve a novel MWCNTs@ZnO/PMAEFc organic–inorganic nanohybrid for modification of amperometric sensing electrodes. By investigating the influence of the system composition on the electrochemical performances, the fabricated modified electrode sensors are anticipated to be capable of improving the electrochemical signals and lowering the LOD, thus providing a simple, rapid, highly-sensitive, and accurate electrochemical approach for detection of aspartame in real samples.

## 2. Materials and Methods

### 2.1. Materials

The original pristine solid powdered MWCNTs, with particle diameter of 10–30 nm, and length of 10–30 μm, were provided by Chengdu Institute of Organic Chemistry, CAS, and dried in an oven at 120 °C for 24 h before use. Ferrocenecarboxylic acid (FCA, 98%), zinc acetate dihydrate (96%, Zn(CH_3_COO)_2_·2H_2_O, and lithium hydroxide monohydrate (≥90%, LiOH·H_2_O) were obtained from J & K Reagent Company, Beijing, China, and used as received. 2-Hydroxyethyl methacrylate (HEMA, 96%) was offered by Zahn Chemical Technology Co., Ltd., Shanghai, China and used dirrectly. 4-Dimethylaminopyridine (DMAP, a highly effective nucleophilic acylation catalyst, 99%) and dicyclohexylcarbodiimide (DCC, 99%) were purchased from Sigma-Aldrich (Shanghai) Trading Co., Ltd., Shanghai, China. Azobisisobutyronitrile (AIBN, 99%) and 4-cyano-4-(dodecylsulfanylthiocarbonyl)sulfanylpentanoic acid (CDSP, 97%) as chain transfer agent (CTA) were supplied by Aladdin (Shanghai, China) Reagent Company, China. Ethanol (98%), ether (98%), and *N*,*N*-dimethylformamide (DMF, 99.5%) were afforded by Sinopharm Chemical Reagent Co., Ltd., Shanghai, China, and used as received. Aspartame (98%) and perfluorinated ion-exchange resin (5% Nafion solution in aliphatic alcohols/water (85–80/15–20, *w/w*) mixture) were furnished by Aladdin (Shanghai) Reagent Company, Shanghai, China. The samples to be detected—including Coke Zero, Sprite Zero, Mango juice, kiwi fruit juice, strawberry juice, and Lipton lipya tea sachet—were acquired from a local supermarket (Xi’an, China).

### 2.2. Methods

MWCNT@ZnO/PMAEFc conducting organic–inorganic nanohybrids were prepared through a two-step reaction procedure, as shown in Scheme 1. ZnO NPs were first prepared and in situ decorated on the surface of MWCNTs as per a previous work reported elsewhere [28,35], and then PMAEFc homopolymers was synthesized through in situ reversible addition-fragmentation chain transfer polymerization (RAFT) and in situ anchored or hybridized on the surface of MWCNTs or MWCNTs@ZnO nanocomposites.

#### 2.2.1. Preparation of MWCNTs@ZnO Nanocomposites

Nanostructured ZnO was prepared using chemical route through reaction of zinc acetate dihydrate with LiOH·H_2_O in alcoholic medium as described previously [28,35,36,37]. Dried MWCNTs (0.20 g) and 1.07 g (4.87 mmol) zinc acetate dihydrate were dispersed and dissolved in 75 mL ethanol by ultrasonication for 2 h in a three-neck flask. The dispersion system was refluxed at 75 °C for 2 h and then cooled to room temperature. After that, the ultrasonically dissolved LiOH·H_2_O of about 0.29 g (6.91 mmol) in 50 mL of ethanol was dropwise added to the flask under vigorous magnetic stirring. The reaction solution was stirred for 3 h to obtain a fine ZnO sol containing MWCNTs. The resulting sol dispersoid was filtered to achieve a solid product mainly containing MWCNTs and ZnO NPs. The solid samples were sintered at 85 °C in vacuum for 2 h to afford MWCNTs@ZnO powder [28,35,36]. The powder sample was dispersed in DMF at a volume ratio of sample/DMF of 1:100 with vigorous stirring for one day to further form covalent interactions. The dispersions were filtrated, and the precipitates were flushed with ethanol twice for easier drying and substitution of DMF. A black solid product MWCNTs@ZnO nanocomposite was obtained by drying in vacuum at 60 °C till constant weight (Yield: 98.2%).

#### 2.2.2. Preparation of MWCNTs@ZnO/PMAEFc Nanohybrids

To prepare MWCNTs@ZnO/PMAEFc organic–inorganic nanohybrids, 2-methacryloyloxyethyl ferrocenecarboxylate (MAEFc) was synthesized via esterification reaction of FCA with HEMA using DMAP as a catalyst and DCC as a dehydration agent in a molar radio of FCA:HEMA:DMAP:DCC of 1:1.4:1:1.2. In a sealed 500 mL three-neck flask, FCA (9.20 g, 40 mmol), HEMA (6.79 mL, 56 mmol), and DMAP (4.89 g, 40 mmol) were dissolved in 50 mL dried CH_2_Cl_2_. Afterwards, DCC (9.9 g, 48 mmol) dissolved in 30 mL desiccative CH_2_Cl_2_ was dropwise added to the above mixed solution under the condition of N_2_ atmosphere at 0 °C at a speed of 3–4 drop s^−1^. The reaction mixture was stirred at 0 °C for 2 h, and then at room temperature overnight. The resulting solution was filtered to get rid of the sediment 1,3-dicyclohexylurea (DCU). The filtrate was extracted twice by using saturated sodium bicarbonate solution and deionized water to remove DMAP and unreacted FCA until the supernatant is colorless. After concentrated, the extract was purified by column chromatography using a mixture of *n*-hexane and ethyl acetate (***v/v*** = 9/1) as an eluent. The collected AEFc solution was evaporated by a rotary evaporator to remove solvents and then dried in vacuum at 30 °C for 24 h, giving an orange solid product with a mean yield of 72%. ^1^H NMR (400 MHz, CDCl_3_), δ (ppm): 4.18 (5H, s, -C_5_H_5_), 4.40 (2H, m, *m*-C_5_H_4_), 4.81 (2H, m, *o*-C_5_H_4_), 4.48 (4H, m, -C(O)OCH_2_-CH_2_OC(O)-), 5.63 (dd, 1H in CH_2_=CCH_3_C(O)OCH_2_-), 6.18 (dd, 1H in CH_2_=CCH_3_C(O)OCH_2_-), and 1.98 (3H, s, -CH_3_).

Then, MWCNTs@ZnO/PMAEFc organic–inorganic nanohybrids were prepared by an in situ RAFT polymerization of MAEFc at a molar ratio of monomer:initiator:CDSP of 100:1:0.25. The mass percentage of the resulting PMAEFc theoretically retained 25%, 50%, and 75% through changing the amount of MWCNTs@ZnO. In a typical procedure,0.17 g dried MWCNTs@ZnO was dispersed in 5 mL DMF under ultrasonication for 2 h in a 25 mL dried Schlenk flask, and then 0.51 g (1.49 mmol) MAEFc, 6.00 mg (0.0149 mmol) CDSP, and 0.61 mg (0.0037 mmol) AIBN were added into the above flask in succession. The reaction system was degassed by freezing, vacuumizing, and purging with N_2_, and the flask was immerged in an oil bath of 75 °C to start the polymerization. After 24 h, the polymerization was terminated by putting the flask into liquid N_2_. The reaction mixture solution was vacuum filtered, and the solid was flushed with DMF thrice to remove the unreacted MAEFc**.** The crude product was precipitated with ether thrice, then filtered, and the solid sample was dried in a vacuum oven at 60 °C overnight, giving a MWCNTs@ZnO/PMAEFc nanohybrid with a PMAEFc mass content of 25% (mean yield: 90.1%), named C1. Similarly, the MWCNTs@ZnO/PMAEFc nanohybrids with a PMAEFc mass percentage of 50% and 75% was prepared in the same method as the above operation, giving a mean yield of 91.8% and 93.4%, respectively, denominated C2 and C3.

### 2.3. Preparation of Modified Electrode Sensors

The above C1, C2, or C3 nanohybrids of 0.01 g were dispersed in 1 mL DMF by means of an ultrasonic technique for 2 h. Uniformly dispersed conducting nanohybrid dispersions (6 µL) were evenly deposited on the surface of the clean pre-treated glass carbon electrodes (GCEs) using a micro-syringe, and then the modified electrodes were dried with an infrared lamp to obtain a finished GCEs coated with the nanohybrids. To ensure good adherence of the nanohybrids on the surface of the electrodes, 5 µL of 1 wt % Nafion solution was cast on the modified layers and dried in air before use. The MWCNTs@ZnO/PMAEFc modified electrode sensors with MAEFc contents of 25%, 50%, and 75% severally denominated as C1, C2, and C3.

### 2.4. Characterization and Measurements

Fourier transformation infrared (FI-IR) spectra were determined by EQUINX55 Fourier transform infrared spectrophotometer (Brucker Corp., Karlsruhe, Germany) using KBr pellets. X-ray diffraction (XRD) analyses were conducted on a D/Max-2550 VB+/PC X-ray diffractometer (Rigaku, Tokyo, Japan) using Cu radiation at a voltage of 40 kV, a current of 30 mA and a scanning rate of 10° min^−1^. Raman spectra were analyzed by an AlMEGA dispersive Raman spectrometer (AlMEGA-TM, Therm Nicolet Corp., Madison, WI, USA) employing Ar^+^ laser with excitation wavelength of 532 nm. Thermal gravimetric analysis (TGA) was performed on a thermoanalyzer system (Q1000DSC + lNCS + FACS Q600SDT, TA Corp., New Castle, DE, USA) at a heating rate of 10 °C min^−1^ under N_2_ atmosphere (flow rate: 40 mL min^−1^). X-ray photoelectron spectroscopy (XPS) spectra were recorded on an AXIS ULTRA spectrometer (Kratos Analytical Ltd., Shimadzu Corp., Kyoto, Japan) at a voltage of 15 kV and room temperature. 

Atomic force microscope (AFM, Dimension ICON, Brucker Corp., Karlsruhe, Germany) was employed to observe surface topography and distribution of MWCNTs@PMAEFc at a scan rate of 0.977 Hz, and to estimate the layer thickness of MWCNTs@PMAEFc. The sample morphologies were recorded on a transmission electron microscope (TEM, JEM-2100, Electron Corp., Osaka, Japan) at an accelerating potential of 200 kV. Before observation, the samples to be determined were ground into powders, and fully dispersed in ethanol through an ultrasonic clearer for about 20 min. After that, a drop of the dispersion solution was dripped on the copper grid with carbon films, and dried for 6 h. Scanning electron microscopy (SEM) images were acquired via a SU-8020 field emission electron microscope (FESEM, Hitachi High-Technologies Corp., Tokyo, Japan) at an operating voltage of 15 kV. All measurements were carried out at room temperature. 

Electrochemical analyses including cyclic voltammetry (CV) and differential pulse voltammetry (DPV) were conducted on a CHI 660E electrochemical workstation (Shanghai Chenhua Co., Shanghai, China) consisting of a conventional three-electrode cell, which contains a bare or modified glassy carbon electrode with a diameter of 3.0 mm as a working electrode, a platinum electrode as a counter electrode, and a saturated calomel electrode (SCE, 0.2415 V vs. SHE (standard hydrogen electrode)) as a reference electrode. For detection of aspartame, DPV analyses were conducted in PBS of pH = 2, with a modulation amplitude of 50 mV, a pulse width of 50 ms and a step potential of 5 mV.

### 2.5. Real Samples Analysis

The two carbonate beverages (Coke Zero and Sprite Zero), three fruit juices (mango juice, kiwi fruit juice, and strawberry juice) and Lipton lipya tea sachet were selected as real sample for residual aspartame detection. For carbonate beverages, their aqueous samples were ultrasonicated for 30 min and diluted with deionized water. Subsequently, the Coke Zero and Sprite Zero were directly spiked with appropriate amounts of aspartame standard solution. Specifically, a 50 mL 1 × 10^−3^ mol L^−1^ aspartame in PBS (pH = 2) solution was prepared, and then the solution of 5 mL was mixed with 100 mL Coke Zero and 100 mL Sprite Zero to get a 14 mg l^−1^ (4.76 × 10^−5^ mol L^−1^) aspartame solution. As for fruit juice, 0.3 mg of each sample was weighed and diluted using 20 mL PBS (pH = 2), whilst Lipton lipya tea sachet (2 g) was fetched out and dissolved in deionized water, and then transferred in a 1 L volumetric flask. For reducing the error, all solutions were filtered via a 0.45 µm membrane before measurements. The aspartame concentration was determined by the electrochemical modified electrodes with the method of DPV. The percentage of recovery was determined for five different concentrations, and relative standard deviations (RSDs) were measured for 20 times detection for the same concentration.

## 3. Results and Discussion

### 3.1. Structural Characterization

The phase structure of MWCNTs@ZnO/PMAEFc nanohybrid and its precursors was examined by XRD, as shown in Figure 1A. The MWCNTs possess characteristic diffraction peaks at 2*θ* of about 25.9°, 43.0°, and 44.2° that are assigned to the diffraction signals of the (002) hexagonal graphite structure, the (100) and (101) diffraction planes, respectively [38]. In XRD patterns of the MWCNTs@ZnO nanocomposites in Figure 1B(b), five strong diffraction peaks emerge at 2*θ* of 31.6°, 34.3°, 36.1°, 47.4°, and 56.5°, which originate from (100), (002), (101), (110), and (102) crystal planes of hexagonal ZnO that matches with the JCPDS card (no. 36-1451) [36,39]. The average crystalline size is estimated to be about 12.61 nm according to Debye–Scherrer’s equation D = *kλ*/βcos*θ*, where k is constant (about 0.9), λ is the wavelength of X-ray (0.15406 nm), β is the full width of half-maxima (FWHM) of the diffraction line corresponding to (101) having high intensity and θ is the Bragg’s angle. This value is in agreement with that estimated by the following SEM and TEM observations. However, the three peaks reflecting the diffraction features of MWCNTs are clearly weakened due to the incorporation of ZnO NPs; but the combination of ZnO NPs with MWCNTs does not destroy the MWCNTs structures. These results show the formation of MWCNTs@ZnO nanocomposites. After PMAEFc is grafted onto the surface of the MWCNTs@ZnO nanocomposites, as shown in Figure 1A(c), a wide and weak peak at 22.3–28.5° is attributed to the overlapping feature of PMAEFc and MWCNTs, and all the five characteristic peaks of ZnO NPs are slightly shifted by about 0.3° in comparison with MWCNTs@ZnO, locating at 31.3°, 34.0°, 35.7°, 47.1°, and 56.2°, probably because of the interactions between ZnO or MWCNts@ZnO and the grafted PMAEFc polymers, indicating the preparation of MWCNTs@ZnO/PMAEFc nanohybrids. 

FTIR spectroscopy was adopted to qualitatively confirm the chemical structure of MWCNTs@ZnO/PMAEFc nanohybrids, as depicted in Figure 1B. For MWCNTs, a wide and strong peak at 3460 cm^−1^ is assigned to the stretch band of the -OH groups due to the moisture absorbed from the air and the residues of carboxylic groups during preparation, and the vibration peak at 1638 cm^−1^ is attributed to the C=C stretch vibration band of the benzenoid (ν_C=C_) structure in the MWCNTs skeleton, signifying the graphite structure of MWCNTs [40,41]. After MWCNTs are decorated with fine ZnO NPs, as illustrated in Figure 1B(b), a new vibration peak emerges at 446 cm^−1^, which is attributable to the Zn-O characteristic peak [42,43]. The peaks at 1643 and 1531 cm^−1^ are ascribed to the characteristic C=C mode, as stated. The increased peak intensity at 3445 cm^−1^ is due to the stretching vibration band of water molecules on the surface of ZnO or MWCNTs/ZnO and MWCNTs [44]. The resulting nanohybrid produces some new vibration peaks attributed to PMAEFc characteristic modes in Figure 1B(c): the vibration band at 3100 cm^−1^ is assigned to the =C-H stretch mode of the ferrocenyl rings (Cp); the bands at 2960–2986 cm^−1^ are attributable to the C-H stretch mode; the peak at 1719 cm^−1^ is attributed to the C=O ester carbonyl stretch characteristic; the peak at 1589 cm^−1^ are assigned to the C=C stretch modes of the Cp rings; the peak at 1138–1276 cm^−1^ is correlated with the C-O stretch mode; and the bands at 772, 828, and 968 cm^−1^ reflect the =C-H stretching modes in the Cp rings and Fe-C or Cp-Fe stretch modes, respectively [45]. The vibration bands at 495 and 455 cm^−1^ are attributed to the Fe-C/Cp-Fe and Zn-O stretching modes.

Raman spectroscopy was further adopted to confirm the preparation of the prepared nanohybrids, as shown in Figure 1A. In all five samples, one can see two obvious peaks at wavenumbers of about 1337 and 1568 cm^−1^, which is known as typical D band (disorder induced band) and G band (graphite band), respectively [46]. The peak intensity ratio I_G_/I_D_, as a rough measure of specimen quality, can be used to indicate the integrity of crystalline graphitic structure of the grown MWCNTs. In comparison with the high I_G_:I_D_ ratio (1.94) of the original pristine MWCNTs, the MWCNTs decorated with ZnO NPs gives a decreased I_G_/I_D_ value of 1.46, signifying an increased interaction between the MWCNTs and ZnO NPs presumably because of the growth of Zn NPs on the defect dots of MWCNTs [47]. After the follow-up RAFT polymerization, the I_G_/I_D_ values further decrease to 1.18, 1.20, and 1.32; this correspondto the MWCNTs@ZnO/PMAEFc nanohybrids with the PMAEFc mass percentage of 25%, 50%, and 75%, respectively. However, the increased I_G_/I_D_ values with increasing the PMAEFc mass percentage are maybe due to more π–π conjugation interactions between MWCNTs and Cp, which have a contribution to good crystalline graphitic structure of the MWCNTs. These findings indicate that the ZnO NPs and PMAEFc homopolymer are decorated or covered at the defect dots or/and on the surface of MWCNTs successively.

Thermogravimetric (TGA) was implemented to describe the thermal behavior of the products in the process from 0 to 800 °C in Figure 2B. About 4.8% of weight loss until 800 °C is put down to the evaporation of absorbed water and ethanol remaining in the original pristine MWCNTs in Figure 2B(a). For MWCNTs@ZnO, the mass loss process can be divided into the following three steps: the first mass loss of ca. 3.7% at 260 °C is probably ascribed to the evaporation of solvents ethanol and DMF in the preparation process. As the temperature is increased from 300 to 720 °C, the inconspicuous weight loss of about 5.8% appears because of removal of oxygenous groups and degradation of residual carbon backbones such as acetate. At more than 720 °C, the weight loss up to around 13.5% corresponds to the pyrolysis or decomposition of the Zn-organic framework [48]. For MWCNTs@ZnO/PMAEFc with PMAEFc 75% (C3), three mass loss steps appear: the mass loss of about 6.2% from 25 to 220 °C results from the evaporation of residual solvents. As the temperature rises up to 450 °C, the mass loss of ca 46.3% is presumably due to the decomposition of the ester bonds and Cp structure of the PMAEFc side chains covered on the surface of the MWCNTs@ZnO nanocomposites. The follow-up mass loss of around 22.9% from 550 to 700 °C is attributed to the carbon–carbon bond degradation of the PMAEFc backbones or main chains. The residual amount of about 24.6% is attributable to the residues of MWCNTs, Zn and Fe atoms. It is clear that at least 69.2% PMAEFc moieties are incorporated in the MWCNTs@ZnO nanocomposites, forming MWCNTs@ZnO/PMAEFc organic–inorganic nanohybrids.

### 3.2. Morphologies of MWCNTs@ZnO/PMAEFc Nanohybrids

Figure 3 shows SEM images of typical MWCNTs, MWCNTs@ZnO, and MWCNTs@ZnO/PMAEFc nanohybrids. Apparently, the as-grown MWCNTs shows a smooth and randomly-entwining surface topology in clusters with some amorphous carbons, and the tube diameters range from about 10 to 35 nm in the inset of Figure 3a, as specified by the manufactory. After in situ decorated with ZnO NPs, it can clearly be noticed that MWCNTs and ZnO NPs coexist in the composites, and low-density and loose ZnO NPs are firmly anchored onto the network structure of MWCNTs, and the particle size of ZnO NPs ranges from 8 nm to 15 nm, forming a shaggy and staggered layered structure consisting of MWCNTs and ZnO NPs, as shown in Figure 3a, which is similar to the topologies reported elsewhere [47]. This close combination contributes to the tight attachment of ZnO NPs on the surface of the electrodes and the enhancement of the electron transfer rate. With the follow-up in situ RAFT polymerization of MAEFc, the morphologies different from MWCNTs and MWCNTs@ZnO can be observed and confirm the co-existence of MWCNTs@ZnO and PMAEFc, as depicted in Figure 3b–d. A low amount of the polymer layers on the surface of the MWCNTs or MWCNTs@ZnO leads to shaggy honeycombed morphologies (Figure 3b), close to the topologies of MWCNTs@ZnO. In the meanwhile, some close-grained granules uniformly scattered and inserted in these mesh-like structures or regions, which is ascribed to the incorporation of PMAEFc matrices. With increasing the addition amount of the polymer layers in Figure 3c, the surface of MWCNTs@ZnO/PMAEFc nanohybrids gradually becomes more compact and uniform, and single and independent PMAEFc granule cannot nearly be observed, forming a blurred interface between CNTs or MWCNTs@ZnO and PMAEFc moieties, and assuming honeycomb morphology. Particularly, in a high content of PMAEFc matrices, as depicted in Figure 3d, the MWCNTs or MWCNTs@ZnO is very closely covered with a thick layer of PMAEFc matrices along the axial direction of the tubes, and the tube diameters significantly increase. As a result, the as-grown MWCNTs can be hardly seen due to the covering of a large mass of PMAEFc matrices, constituting a unique morphology that appears to consist of closely connected PMAEFc particles. These different characteristics of morphologies are consistent with the growth of ZnO NPs and different covering amounts of PMAEFc matrices on the surface of the MWCNTs. The different combination of the nanohybrids is anticipated to behave differently in their electrochemical process and is thus used for tuning different electrochemical properties.

TEM observations were further used to characterize the morphologies of MWCNTs@ZnO and MWCNTs@ZnO/PMAEFc, as presented in Figure 4. The as-grown MWCNTs are observed to have different outer diameters between ca 7.5 and 20 nm, and their wall surface is smooth and tangled with each other, without any detectable amorphous layer excluding several blobs of amorphous carbons. The TEM image of MWCNTs@ZnO nanocomposites shows some spherical nanograins with particle sizes of about 7–10 nm, which take on a slightly dark color, are scattered on the surface or the side walls of MWCNTs, constructing a mesh-like structure. This is characteristics of ZnO NPs, suggesting the decoration of ZnO NPs on the surface of MWCNTs or preparation of MWCNTs@ZnO nanocomposites. After the in situ RAFT polymerization of MAEFc, as illustrated in Figure 4c,d, the blurred interfaces for CNTs appear, the roughness of the surface increases, and the size of the CNTs is widened at different levels, and the maximal size is about 32 nm for C2 and 35 nm for C3. This suggests that PMAEFc polymer matrix layers with different mass proportions are decorated on the surface of MWCNTs or MWCNTs@ZnO. In the meanwhile, MWCNTs are closely surrounded with the above latticed structure constructed by ZnO NPs. The rough and porous topologies endow the MWCNTs@ZnO/PMAEFc nanohybrids with a high surface-to-volume ratio, which makes for the improvement of the electrochemical response.

### 3.3. Electrochemical Characteristics

CV was adopted to examine the electrochemical behavior of various materials, as displayed in Figure 5A. It can be noticed that the bare GCE and the modified GCE electrodes coated with both MWCNTs and MWCNTs/PMAEFc fail to produce any obvious redox response in PBS solution of pH = 2; whereas the MWCNTs@ZnO ones generate slightly high current response compared with the above electrodes. Clearly, the data indicate that the incorporation of ZnO NPs seems to be able to slightly improve the sensitivity. However, the modified electrodes based on only any single material, even the simple combination of two components including MWCNTs, PMAEFc, and ZnO NPs surprisingly do not show well-defined redox peak currents, which is different from the results reported by quite a lot of references in that ferrocene based compounds with CNTs modified electrodes have shown excellent ferrocene peaks [49,50,51]. It has been reported that the electrical properties of the composites depend on the aspect ratio, alignment, and even alignment thickness, and dispersion of conductive fillers, and the alignment of MWCNTs in a certain way, viz. the alignment direction, is conductive; or else, the MWCNTs will be insulating [52,53]. Therefore, the reason why no redox peak is observed for MWCNTs and MWCNTs@PMAEFc is probably due to the random and nondirectional alignment of MWCNTs without the field, as demonstrated in Figure 3 and Figure 4. Moreover, the one-dimensional disorderly aligned MWCNTs have a low layer thickness. These factors result in poor conducting properties of MWCNTs and ferrocene-containing MWCNTs@PMAEFc nanocomposites.

XPS and AFM were further used to interpret the electrochemical response, as illustrated in Figure 5B,C. XPS survey spectra in Figure 5B show the surface elemental compositions of the sample to be determined. MWCNTs produce two characteristic photoelectronic signals at binding energies (BEs) of 285.3 and 533.1 eV assigned to the C 1s and O 1s signals, respectively. The appearance of Fe 2p at 708.5–721.8 eV and Fe 3p peaks at 53.4 eV in Figure 5B(b) indicates that the ferrocene-containing polymer moieties were covered on the surface of the MWCNTs, forming MWCNTs/PMAEFc, with the Fe mass content of about 8.6%. The AFM image in Figure 5C clearly reveals that the MWCNTs@PMAEFc forms a randomly-aligned, chaotic and aggregated morphology with a mesh-like structure, and a total layer thickness is about 48–69 nm. The thickness of PMAEFc layers is estimated to be ca. 18–39 nm based on the size supplied by SEM and TEM as well as producer; these results are in accordance with those by TEM and SEM. The MWCNTs@PMAEFc nanocomposites with this kind of randomly-arranged morphology, as stated above, would lead to poor conductivity and thus poor electrochemical redox response [52,53]. On the other hand, in comparison with the bulk PMAEFc moieties, the relatively low PMAEFc composition based on XPS result and layer thickness in MWCNTs@PMAEFc is maybe also responsible for the poor current response. In contrast, the modified MWCNTs@ZnO/PMAEFc/GCE electrodes produce well-defined and quasi-reversible redox peak currents, with peak potential differences or peak separations ∆*E* (∆*Ep* = *Ep*,anodic − *Ep*,cathodic) lower than about 50 mV and the ratio of the anodic to cathodic peak currents (I*_p_*_,a_/I*_p_*_,c_) almost close to unity 1, which is due to the reversible Fe(III)/Fe(II) redox electrochemical process [54]. Consequently, it is concluded that the synergistic effect among the components may be responsible for good electrochemical response performances, and the C1, C2, and C3 modified electrodes are reasonably selected for the following analysis including aspartame detection. As the component proportion of PMAEFc is increased from C1 to C3, the redox peak currents of the modified electrodes are enhanced in that more MAEFc units are oxidized or more MAEFc+ are reduced [55]. These results signify that the prepared MWCNTs@ZnO/PMAEFc nanohybrids can potentially be fabricated into an electrochemical modified electrode with good redox reversibility and can be tuned by altering the system compositions.

Given that the scan rate may have effect on the electroactivity of the nanohybrids, viz., the conducting rate of the electrons, the peak currents (*I*_p_) dependence on the scan rates was examined in a potential range from 10 to 120 mV s^−1^ at a voltage step of 10 mV in PBS solution of pH = 2.0. Figure 5D clearly shows the peak currents are in direct proportion to the scan rate in a rate range from 10 to 120 mV s^−1^, showing a good linear correlation between the scan rates and peak currents, and offering a well-defined linear equation *I*_p,a_(10^−4^ A) = 0.0206ν (mV s^−1^) + 0.0016 (*R*^2^ = 0.9925) for anodes and *I*_p,c_(10^−4^ A) = −0.0223ν (mV s**^−1^**) − 0.0021 (*R*^2^ = 0.9962) for cathodes. These findings suggest that the redox processes are a diffusion-controlled electrode reaction process for the CNTs@ZnONPs/PMAEFc modified electrode sensors and the electrode processes are quasi-reversible [56,57]. The unique electrochemical traits of the resulting nanohybrids are expected to be used to detect trace additives in food or beverages.

### 3.4. Applications of the Modified Electrochemical Sensors

Albeit the electrochemical sensors have offered a highly-efficient, fast, accurate method of detecting the trace additives in food and beverage, the key issues that people concern are their long-term stability and/or lifetime during their use. For this, the repeatability and stability of the MWCNTs@ZnO/PMAEFc modified electrodes are examined in PBS of pH 2.0 at a constant aspartame concentration of 1 × 10^−8^ mol L^−1^. For the durability or stability of the electrochemical sensor, variation in DPV peak currents with time during storage in air for consecutive fifteen days was investigated using the same freshly prepared representative C2 modified electrode, as shown in Figure 6A. Clearly, the modified C2 electrode produces fairly small change in current responses, and retains about 99% of its initial value after stored for 15 days, and the current response only deviates by 1.14%, signifying a significantly high stability or durability. Consequently, the sensing device possesses high use efficiency. In the case of the same aspartame concentration, the reproducibility of the same modified electrode was evaluated through five replicate measurements in both Coke Zero and Sprite Zero, as shown in Figure 6B,C. The relative standard deviation (RSD) is determined to be only 2.0 for Coke Zero and 3.1% for Sprite Zero, indicating a good reproducibility. The electrode-to-electrode repeatability was evaluated by six modified electrodes prepared individually in the same way in 1 × 10^−6^ mol L^−1^ aspartame solution using DPV, as illustrated in Figure 6D. The RSD of 1.7% is achieved suggesting the high precision and/or reproducibility of the proposed procedure for detection of aspartame.

### 3.5. Quantitative Detection Analysis of Aspartame

To practically measure trace aspartame in food or beverages, calibration curves of the nanohybrid modified electrode sensors should be prepared by detecting the change of peak currents with the concentrations of aspartame using DPV methods. The detection was conducted at room temperature in a traditional electrochemical cell having 10 mL 0.2 mol L^−1^ PBS buffer solution (pH = 2.0) by changing the aspartame content. The experimental results were shown in Figure 7. It is clear that the anodic or oxidation peak currents increase linearly with increasing the concentrations of aspartame in a wide range from 1 × 10^−8^ to 1 × 10^−12^ mol L^−1^, and the increment in peak currents (I-I_0_) is proportional to the logarithm of aspartame concentrations, giving a linear calibration equation *I*_p_,_C1_(1 × 10^−5^ A) = −0.0354(−log C) − 0.3303 (*R*^2^ = 0.9982, *n* = 7) for C1 and *I*_p,C2_(1 × 10^−5^ A) = −0.0960(−log C) − 0.1080 (*R*^2^ = 0.9880, *n* = 7) for C2, *I*_p,C3_(1 × 10^−5^ A) = −0.0483(−log C) − 0.2232 (*R*^2^ = 0.9948, *n* = 7) for C3, where *I*_p_ and C represent the peak current and the concentration of aspartame, respectively. The limit of detection (*LOD*) is calculated as per the following equation using three times noise, viz., signal/noise = 3 (*S*/*N* = 3) [58]
*LOD* = *KS*/*r*(1)
where S is the standard deviation of signals measured 20 times for blank solution, and it is about 3.682 × 10^−10^ A for C1, 4.320 × 10^−11^ A for C2 and 1.547 × 10^−9^ A for C3; *K* is a coefficient based on a certain confidence level factor (*K* = 3); and r is the sensitivity of the detection method, viz., the slope of the calibration curve (Here, *r* = 0.0354, 0.0960 and 0.0483 A M^−1^ for C1, C2 and C3, respectively). The LOD values are estimated to approximately be 3.12 × 10^−8^, 1.35 × 10^−9^ and 9.61 × 10^−8^ mol L^−1^ corresponding to the C1, C2, and C3 modified electrode sensor, respectively. These LOD values signify that an effective combination of the three constituents can significantly lower the LOD value of the modified electrodes, and thus detect an extremely trace amount of additives in food or beverage. These values are superior to those obtained from any other technique such as flow injection analysis (FIA), surface-bound molecular imprinting technology (SMIT), capillary zone electrophoresis (CZE), square-wave voltammetry (SWV), FTIR, DPV, hydrophilic interaction liquid chromatography (HILIC), and UV visible simultaneous analysis (CZE-UV) [10,28,59,60,61,62,63,64,65,66,67,68], and also lower than the result only obtained from MWCNTs@ZnO modified electrode sensors [28], as summarized in Table 1. This may be ascribed to unique redox electrochemical properties and fast electron transfer feature of ferrocene groups at the electrode interfaces based on a synergetic effect of these constituents [29,30]. In contrast, under the same conditions, the aspartame detection with both the nonmodified electrodes and the PMAEFc modified electrodes offers poor DPV curves, with no peak current, as shown in Figure 7D, which produces irregular nonlinear or nonproportional calibration equations, as expected (not shown). Therefore, the detection based on the nonmodified and PMAEFc modified electrodes is low-reliability and meaningless compared with their C1, C2, and C3 counterparts, whereas the proposed approach is advantageous for the aspartame detection. Based on these findings, it is inferred that the developed modified electrode sensors exhibit remarkable advantages, with high-sensitivity responsivity, low LOD, and wide linear detection range, and can be used to detect trace food additives.

To evaluate the applicability of the MWCNTs@ZnO/PMAEFc/GCE sensors, the C2 sensor was selected to determine the aspartame concentrations in the real samples. For this purpose, the Coke Zero, Sprite Zero, mango juice, kiwi fruit juice, strawberry juice, and Lipton lipya tea sachet sample solutions of 50 mL, which contain aspartame, were stirred for 30 min and served as stock solutions with no dilution. Then, the solution was added to the sample cells and the alteration in current density was detected by DPV. The electrochemical signals of each sample were recorded five times and averaged, with the RSD as low as 0.67%, 0.17%, 2.6%, 1.8%, 3.3%, and 4.1% for Coke Zero, Sprite Zero, mango juice, kiwi fruit juice, strawberry juice, and Lipton lipya tea sachet samples, respectively. The concentrations of aspartame in Coke Zero, Sprite Zero, mango juice, kiwi fruit juice, strawberry juice and Lipton lipya tea sachet were calculated using the calibration graph to be approximately 35.36 µM, 40.20 µM, 0.96 mM, 1.34 mM, 0.88 mM, and 1.53 mM, respectively in this way. Compared with any other method presented in the literatures (Table 1) [28,60,61,62,63,64,65,66,67,68], this approach offers significantly low RSD and can expand the detection of the aspartame in real samples. Consequently, the developed MWCNTs@ZnO/PMAEFc/GCE modified electrode sensors are proved to be an effective tool for quick and simple detection of aspartame contents in real beverage or food samples.

## 4. Conclusions

In summary, we have shown an effective means to prepare novel organic–inorganic nanohybrids constructed by carbon nanotube hybridized with ZnO NPs and wrapped with PMAEFc homopolymer, as confirmed by FTIR, Raman, XRD, TGA, XPS, AFM, TEM, and SEM measurements. The MWCNTs@ZnO/PMAEFc nanohybrid modified electrodes exhibit a quasi-reversible electrochemical redox response, reproducibility, stability, and a good linear correlation between the oxidation peak current and the aspartame concentration over a wide concentration range, hinging on the effective combination of the three components, and/or their composition ratios. A significantly low LOD of 1.35 × 10^−9^ mol L^−1^ can be applied to detect trace aspartame in a μM concentration level in food or beverage, particularly for sample C2. Compared with previously-reported techniques, the developed electrochemical sensors have the advantages of good electrochemical response, repeatedly stability with low RSD, low LOD, and wide detection range. Therefore, the prepared MWCNTs@ZnO/PMAEFc nanohybrid modified electrode electrochemical sensors can potentially applied in the monitoring of beverage and food safety.

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
