# Peer review of "Organic–Inorganic Nanohybrid Electrochemical Sensors from Multi-Walled Carbon Nanotubes Decorated with Zinc Oxide Nanoparticles and In-Situ Wrapped with Poly(2-methacryloyloxyethyl ferrocenecarboxylate) for Detection of the Content of Food Additives"

_nanomaterials, 2019, doi:10.3390/nano9101388_

Round 1

Reviewer 1 Report

The authors present a new material composite to detect aspartame. The work is clean and to the point but I am not 100% convinced if it is novel enough. The method has been reported before which has also been cited by the authors. Having said that, the authors should demonstrate the key experiment as described in point 1 to truly show that the composite is better than what has been reported so far?

The authors did not present any data on why/how ZnOis improving the sensitivity. Have the authors compared the same composite material without ZnO? The authors should compare the performance between (a) MWCNTs@ZnO/PMAEFc and (b)MWCNTs/PMAEFc. This will prove if ZnO is useful or not.  Authors mentioned about nanoparticles: Could you the authors address why only ZnO and not gold nanoparticles, silver nanoparticles, reduced graphene?  Was there a specific reason for using glassy carbon electrode because as the authors describe the application being industrial which needs to be cost-effective, why not use screen printed electrodes which will have an enlarged surface and will also help to get even better sensitivity.  In line 66, the authors mentioned about some issues of biosensors? Could the author be specific to the work being done for aspartame detection, what is the current problem they are trying to address? The authors should present a table discussing different techniques used to measure aspartame and its LOD so that they all can be compared to the current system the authors have developed? What does the author mean by hybridization of ZnO with MWCNTs? lines 77-81 are confusing. The authors should rephrase them so that readers can better understand the point the authors are trying to make.

Author Response

Manuscript ID: nanomaterials-591287

Title: Organic-inorganic nanohybrid electrochemical sensors from multi-walled carbon nanotubes decorated with zinc oxide nanoparticles and in-situ wrapped with poly(2-methacryloyloxyethyl ferrocenecarboxylate) for detection of the content of food additives

Authors: Jing-Wen Xu*, Zhuo-Miao Cui, Zhan-Qing Liu, Feng Xu, Ya-Shao Chen, Yan-Ling Luo*

 Response to reviewer #1

The authors present a new material composite to detect aspartame. The work is clean and to the point but I am not 100% convinced if it is novel enough. The method has been reported before which has also been cited by the authors. Having said that, the authors should demonstrate the key experiment as described in point 1 to truly show that the composite is better than what has been reported so far?

► We are thankful for your valuable comments. Based on your comment, we have carefully checked and revised our manuscript including English language and style. We wish that the revised manuscript will be satisfactory, and finally be accepted for publication in Nanomaterials.

The authors did not present any data on why/how ZnO is improving the sensitivity. Have the authors compared the same composite material without ZnO? The authors should compare the performance between (a) MWCNTs@ZnO/PMAEFc and (b) MWCNTs/PMAEFcThis will prove if ZnO is useful or not.

► We appreciate your valuable comment, which will help improve the quality of our manuscript. To confirm why/how ZnO improves the sensitivity, we have measured the CV curve of MWCNTs/PMAEFc composites, and compare the CV curves of MWCNTs, MWCNTs/PMAEFc, MWCNTs@ZnO and MWCNTs@ZnO/PMAEFc. We notice that the MWCNTs, MWCNTs/PMAEFc modified electrodes produce no current response, whereas the MWCNTs@ZnO electrodes generate slightly high current. Clearly, the data indicate that the incorporation of ZnO NPs seems to be able to slightly improve the sensitivity. However, the modified electrodes based on only these materials do not show well-defined redox peak current. In contrast, the MWCNTs@ZnO/PMAEFc modified electrodes process more obvious redox peak currents. Consequently, it is concluded that their synergistic effect may be responsible for good electrochemical response performances. These results have been incorporated in our revised manuscript.

 Pages 10 and 11, lines 392-406: “It can be noticed that the bare GCE and the modified GCE electrodes coated with both MWCNTs and MWCNTs/PMAEFc fail to produce any obvious redox response in PBS solution of pH=2; whereas the MWCNTs@ZnO ones generate slightly high current response compared with the above electrodes. Clearly, the data indicate that the incorporation of ZnO NPs seems to be able to slightly improve the sensitivity. However, the modified electrodes based on only any material including MWCNTs, PMAEFc and ZnO NPs do not show well-defined redox peak currents. In contrast, the modified MWCNTs@ZnO/PMAEFc/GCE electrodes produce well-defined and quasi-reversible redox peak currents, with peak potential differences or peak separations ∆E (∆Ep=Ep,anodic-Ep,cathodic) lower than about 50 mV and the ratio of the anodic to cathodic peak currents (Ip,a/Ip,c) almost close to unity 1, which is due to the reversible Fe(III)/Fe(II) redox electrochemical process [50]. Consequently, it is concluded that the synergistic effect among the components may be responsible for good electrochemical response performances, and the C1, C2 and C3 modified electrodes are reasonably selected for the following analysis including aspartame detection.”.

Authors mentioned about nanoparticles: Could you the authors address why only ZnO and not gold nanoparticles, silver nanoparticles, reduced graphene?

► We speak highly of your kind comment. According to your comment, we have added gold nanoparticles, silver nanoparticles, reduced graphene in our manuscript. Thank you!

Page 2, lines 71-84: “On the other hand, MWCNTs can be modified and functionalized by anchoring some nanoparticles (NPs) such as zinc oxide (ZnO), gold and silver nanoparticles, reduced graphene on their surface or their external walls [15-18]. These nanomaterials are widely used to fabricate electrochemical sensors due to their excellent electrochemical properties. They can enlarge the electrode superficial area, further increase the electron transfer rate, and ultimately improve the sensitivity of the electrochemical (bio)sensors [15-20]. It has been proved that the electrochemical (bio)sensors, particularly sensors based on screen printed technology, are highly sensitive, easily designable and cost-effective, and readily-miniaturized [12,21-23], but some issues remain to be improved. It is still an important and challenged work for us how to achieve a better electrochemical response and repeatedly stability, minimum LOD and wider detection range for development of electrochemical sensors, whereas selecting and optimizing sensing materials with unique comprehensive performances may be one of the potentially effective routes.”.

Was there a specific reason for using glassy carbon electrode because as the authors describe the application being industrial which needs to be cost-effective, why not use screen printed electrodes which will have an enlarged surface and will also help to get even better sensitivity.

You have proposed a good comment, which is very valuable for industrial applications that need to be cost-effective, and screen-printing electrodes can meet this requirement. It is known that screen-printing is one of the most promising approaches towards simple, rapid and inexpensive production of electrodes or (bio)sensors and it is particularly suited to the mass production of low-cost disposable electrodes or (bio)sensors. One of the most prominent commercialized applications of screen-printed electrodes is the glucose biosensor used for diabetes. Screen printed electrodes have an enlarged surface and help to get better sensitivity. These are a good inspiration to our future study. However, our aim or the emphasis of this study is to compare the electrochemical response of different materials but not the action of the electrodes themselves, the ordinary three-electrode system can be competent for this work. Of course, in the future we would consider adopting the screen printed electrodes so as to obtain fast, economical, reliable, versatile and higher-sensitivity electrochemical sensors or modified electrodes for food safety detection. Thank you

In line 66, the authors mentioned about some issues of biosensors? Could the author be specific to the work being done for aspartame detection, what is the current problem they are trying to address?

Thank you very much for your kind comment. To effectively detect trace aspartame, we expect to develop an electrochemical sensor with good electrochemical response, low limit of detection, wide detection range and repeatedly stability with low RSD, which is our main objective. Consequently, albeit the related materials can be fabricated into biosensors, and their properties can be used for reference in fabricating electrochemical sensors for detection of residual additives in food and/or beverage, some issues of biosensors are weakened in our revised manuscript in line with your comment. In fact, some biosensors can be used for detection of food additives, which have reference significance for us to design and develop novel sensing materials with excellent electrochemical properties [Radulescu M. C.; Bucur B.; Bucur M. P.; Radu G. L. Bienzymatic Biosensors for Rapid Detection of Aspartame by Fow Injection Analysis. Sensors 2014, 14, 1028-1038; Gurusamy M.; Velusamy P.; Chinnachamy C.;  Anand T. P.; Vinurajkumar S. Applications of Biosensors in Food Industry. Biosci. Biotechnol. Res. Asia 10(2):711-714.].

 Page 2, lines 68-84: “Since carbon nanotubes (CNTs) exhibit the activity of edge-plane-like graphite sites at the CNTs ends, they can promote the electron transfer in electroanalytical applications [13], and the electrochemical (bio)sensors based on MWCNTs exhibit high sensitivity and low LOD for food safety detection or quality control [14,15]. On the other hand, MWCNTs can be modified and functionalized by anchoring some nanoparticles (NPs) such as zinc oxide (ZnO), gold and silver nanoparticles, reduced graphene on their surface or their external walls [15-18]. These nanomaterials are widely used to fabricate electrochemical sensors due to their excellent electrochemical properties. They can enlarge the electrode superficial area, further increase the electron transfer rate, and ultimately improve the sensitivity of the electrochemical (bio)sensors [15-20]. It has been proved that the electrochemical (bio)sensors, particularly sensors based on screen printed technology, are highly sensitive, easily designable and cost-effective, and readily-miniaturized [12,21-23], but some issues remain to be improved. It is still an important and challenged work for us how to achieve a better electrochemical response and repeatedly stability, minimum LOD and wider detection range for development of electrochemical sensors, whereas selecting and optimizing sensing materials with unique comprehensive performances may be one of the potentially effective routes.”.

The authors should present a table discussing different techniques used to measure aspartame and its LOD so that they all can be compared to the current system the authors have developed?

► We speak highly of your kind comment, which is very valuable for improving the quality of our article. According to your comment, we have compared the current system we have developed with techniques reported by some literatures, and further discussed these different techniques and the LOD values obtained, as listed in Table 1 in our revised manuscript. Thank you again!

Page 13, lines 480-464: “The LOD values are estimated to approximately be 3.12×10-8, 1.35×10-9 and 9.61×10-8 mol L-1 corresponding to the C1, C2 and C3 modified electrode sensor, respectively. These LOD values signify that an effective combination of the three constituents can significantly lower the LOD value of the modified electrodes, and thus detect an extremely trace amount of additives in food or beverage. These values are superior to those obtained from any other technique such as flow injection analysis (FIA), surface-bound molecular imprinting technology (SMIT), capillary zone electrophoresis (CZE), square-wave voltammetry (SWV), FTIR, DPV, hydrophilic interaction liquid chromatography (HILIC) and UV visible simultaneous analysis (CZE-UV) [10,28,55-64], and also lower than the result only obtained from MWCNTs@ZnO modified electrode sensors [28], as summaried in Table 1. This may be ascribed to unique redox electrochemical properties and fast electron transfer feature of ferrocene groups at the electrode interfaces based on a synergetic effect of these constituents [29,30].”.

What does the author mean by hybridization of ZnO with MWCNTs? lines 77-81 are confusing. The authors should rephrase them so that readers can better understand the point the authors are trying to make.

► We appreciate your valuable comment. In fact, what is the hybridization of ZnO with MWCNTs refers to a composite of ZnO NPs with MWCNTs formed through some interaction including physical mixing, finally forming a hybrid material, which concretely involves in the preparation of ZnO NPs in the presence of MWCNTs, as described in section “2.2.1 Preparation of MWCNTs@ZnO nanocomposites”. This statement of hybridization of ZnO with MWCNTs is also proposed in reference (28. Balgobind K. et al, J. Electroanal. Chem. 2016, 774, 51–57: the original reference number 22 has been changed into 28). In our manuscript, in the case of preparation of MWCNTs@ZnO, the definition of MWCNTs@ZnO nanocomposites is adopted, as described above (2.2.1 Preparation of MWCNTs@ZnO nanocomposites). Thank you!

Page 2, lines 92-96: “It was reported that ZnO NPs could be hybridized with MWCNTs to fabricate electrochemical sensors for improving the electrochemical signal, lowering LOD, and determining aspartame in food and beverage samples [28].”.

With respect to the confused statement in lines 77-81, we have rephrased this paragraph so that readers can better understand the point we are trying to make.

Page 3, lines 96-109: “Ferrocene and its derivatives have attracted significant interest in the area of electrochemical (bio)sensing, semiconducting materials and electrocatalysis etc, due to their excellent electrical activities, fast electron transfer properties and good electrochemical response [29-32]. However, the use of any single component above mentioned does not achieve the expexted effect in our study. The designability and anisotropy of the properties of composite materials, especially their combined effect, including unique additivity, productness, modularity, and seepage behavior [33] inspire us to effectively combine these components to construct a novel hybrid with the enhanced electron transfer rate, the improved electrochemical properties and the minimized LOD value. ”.

Author Response

Manuscript ID: nanomaterials-591287

Title: Organic-inorganic nanohybrid electrochemical sensors from multi-walled carbon nanotubes decorated with zinc oxide nanoparticles and in-situ wrapped with poly(2-methacryloyloxyethyl ferrocenecarboxylate) for detection of the content of food additives

Authors: Jing-Wen Xu*, Zhuo-Miao Cui, Zhan-Qing Liu, Feng Xu, Ya-Shao Chen, Yan-Ling Luo*

 Response to reviewer #2

The Authors developed novel organic-inorganic nanohybrid electrochemical sensor for food additives detection. In recent times, an impressive number of inventive designs and technological advances for sensor development made that greatly contribute to their potential practical applications. After decades of development, sensors and biosensors have been demonstrated as prospective powerful tools for food analysis. The authors fabricated modified electrode sensors suitable for detection of aspartame in real samples. The article is analogous to previous study concerning sensor with gold nanoparticles (DOI: 10.1039/c8tc05294h). The authors strongly relied on formerly established protocols. Characteristic of sensor metrological parameters was well prepared. The manuscript is valid for publication after minor revision.

► We are grateful for your kind comments on our manuscript. According to your comments, we have made corresponding revisions or corrections on our manuscript including minor spell errors. We wish that the revised paper will be satisfactory and finally be accepted by Nanomaterials.

There are still some points that the authors should pay attention to:

The authors should revised manuscript according to Guide for authors attached in journal (eg. Tables, figures captions, etc.). Also, all Latin phrases (via, i.e., e.g., in situ, etc…) in scientific writing should be in italics and abbreviations should be explained while using for the first time. Moreover, lines 69, 70 and other have different text formatting.

► Thank you for your kind comments. In line with your comment, we have revised our manuscript including tables, figures captions, Latin phrases, abbreviations and text formatting throughout the text. See our revised manuscript. Thanks again!

I think it’s a walking on a thin ice claiming that aspartame is toxic for mammals. Especially based on outdated studies [7,8] and studies published in low-reputation journals ([9] - Pakistan Journal of Biological Sciences). More reliable data should be presented or statements about aspartame should be rephrased.

► We are sorry for our inappropriate expression. To avoid the issue, we have deleted the related sentences. Thank you again.

Line 52, Electronic tongues should be taken into consideration as techniques gaining more and more applicability in sweeteners analysis, e.g.

- Evaluation of taste-masking effects of pharmaceutical sweeteners with an electronic tongue system, doi: 10.3109/03639045.2012.758636

- Critical review of electronic nose and tongue prospects in pharmaceutical analysis, doi: 10.1016/j.aca.2019.05.024

► We appreciate your valuable comment, which helps improve the quality of our manuscript. According to your comment, we have added the related description in our revised manuscript.

 Page 2, lines 61-64: “Electronic tongues and electronic noses have received increasing attention as these techniques gain more and more applicability in pharmaceutical industry including sweeteners analysis [11,12]. Nevertheless, their sensitivity, specificity and response range are yet to be improved.”.

Line 64. References proving the statement about biosensors advantages should be presented.

► Thank you very much for your valuable comment. We have provided related references to prove the advantages of biosensors.

 Page 2, lines 77-80: “It has been proved that the electrochemical (bio)sensors, particularly sensors based on screen printed technology, are highly sensitive, easily designable and cost-effective, and readily miniaturized [12,21-23], but some issues remain to be improved.”.

Wasilewski T.; Migon D.; Gębicki J.; Kamysz W. Critical Review of Electronic Nose and Tongue Instruments Prospects in Pharmaceutical Analysis. Analyt. Chim. Acta 2019, 1077, 14–29.

  1. Shobha Jeykumari D. R.; Raman K.; Narayanan S. Nanobiocomposite Electrochemical Biosensor Utilizing Synergic Action of Neutral Red Functionalized Carbon Nanotubes. Micro Nano Lett. 2012, 4, 220–227.
  2. Huang Y.; Xu J.; Liu J.; Wang X.; Chen B. Disease-Related Detection with Electrochemical Biosensors: A Review. Sensors 2017, 17, 2375.
  3. Krishnan S. K.; Singh  E.; Singh  P.; Meyyappan  M.; Nalwa H. S. A Review on Graphene-Based Nanocomposites for Electrochemical and Fluorescent Biosensors. RSC Adv. 2019, 9, 8778–8881.

Line 97, MWCNTs was a solution, suspension or something else before dewatering. What was the concentration then?

► Thank you for your kind comment. In fact, the dewatering process is a drying operation. The MWCNTs here are the original pristine solid powdered ones rather than a solution or suspension. For clarity, we have revised this sentence as follows.

 Page 3, lines 123-125: “The original pristine solid powdered MWCNTs, with particle diameter of 10-30 nm, and length of 10–30 μm, were provided by Chengdu Institute of Organic Chemistry, CAS, and dried in an oven at 120 oC for 24 h before use.”.

Line 98, for how long MWCNTs were dewatering.

► Many thanks for your kind comment. The dewatering time for MWCNTs was 24 h, which has been incorporated in our revised manuscript.

 Page 3, lines 123-125: “The original pristine solid powdered MWCNTs, with particle diameter of 10-30 nm, and length of 10–30 μm, were provided by Chengdu Institute of Organic Chemistry, CAS, and dried in an oven at 120 oC for 24 h before use.”.

More detailed characteristic of FESEM images need to presented.

► Thank you for your kind comment, which is valuable for improving the quality of our manuscript. Based on you comment, we have presented the detailedly characteristic of FESEM images.

 Pages 8 and 9, Line 340-367: “After in situ decorated with ZnO NPs, it can clearly be noticed that MWCNTs and ZnO NPs coexist in the composites, and low-density and loose ZnO NPs are firmly anchored onto the network structure of MWCNTs, and the particle size of ZnO NPs ranges from 8 nm to 15 nm, forming a shaggy and staggered layered structure consisting of MWCNTs and ZnO NPs, as shown in Fig. 3(a), which is similar to the topologies reported elsewhere [49]. This close combination contributes to the tight attachment of ZnO NPs on the surface of the electrodes and the enhancement of the electron transfer rate. With the fellow-up in situ RAFT polymerization of MAEFc, the morphologies different from MWCNTs and MWCNTs@ZnO can be observed and confirm the co-existence of MWCNTs@ZnO and PMAEFc, as depicted in Fig. 3(b-d). A low amount of the polymer layers on the surface of the MWCNTs or MWCNTs@ZnO leads to shaggy honeycombed morphologies (Fig. 3(b)), close to the topologies of MWCNTs@ZnO. In the meanwhile, some close-grained granules uniformly scattered and inserted in these mesh-like structures or regions, which is ascribed to the incorporation of PMAEFc matrices. With increasing the addition amount of the polymer layers in Fig. 3(c), the surface of MWCNTs@ZnO/PMAEFc nanohybrids gradually becomes more compact and uniform, and single and independent PMAEFc granule cannot nearly observed, forming a blurred interface between CNTs or MWCNTs@ZnO and PMAEFc moieties, and assuming honeycomb morphology. Particularly, in a high content of PMAEFc matrices, as depicted in Figs. 3(d), the MWCNTs or MWCNTs@ZnO is very closely covered with a thick layer of PMAEFc matrices along the axial direction of the tubes, and the tube diameters significantly increase. As a result, the as-grown MWCNTs can be hardly seen due to the covering of a large mass of PMAEFc matrices, constituting a unique morphology that appears to consist of closely connected PMAEFc particles. These different characteristics of morphologies are in consistent with the growth of ZnO NPs and different covering amounts of PMAEFc matrices on the surface of the MWCNTs. The different combination of the nanohybrids is anticipated to differently behave in their electrochemical process and thus used for tuning different electrochemical properties.”.

Conclusion section need to be supplemented by comprehensive comparison with previously developed, similar sensors.

► Thank you for your kind comment. We have supplemented the “Conclusion” section by comparing this study with previously developed similar sensors as follows.

 In “Conclusion”: “Compared with previously-reported techniques, the developed electrochemical sensors have the advantages of good electrochemical response, repeatedly stability with low RSD, low LOD and wide detection range. Therefore, the prepared MWCNTs@ZnO/PMAEFc nanohybrid modified electrode electrochemical sensors can potentially applied in the monitoring of beverage and food safety.”.

In the experimental section, blank measurement (with no aspartame) should be added and multiple measurements for each concentration (error bars) should be included. The details on the calculation of the LOD should be added.

► You proposed a good comment, which will promote the quality of our manuscript. As per your comment, we have added the blank experiments and given error bars based on seven measurements (n=7). The details on the calculation of LOD have also been added in our revised manuscript.

Page 13, lines 462-482: “It is clear that the anodic or oxidation peak currents increase linearly with increasing the concentrations of aspartame in a wide range from 1×10-8 to 1×10-12 mol L-1, and the increment in peak currents (I-I0) is proportional to the logarithm of aspartame concentrations, giving a linear calibration equation Ip,C1(1×10-5A)=-0.0354(-log C)-0.3303 (R2=0.9982, n=7) for C1 and Ip,C2(1×10-5A)=-0.0960(-log C)-0.1080 (R2=0.9880, n=7) for C2, Ip,C3(1×10-5A)=-0.0483(-log C)-0.2232 (R2=0.9948, n=7) for C3, where Ip and C represent the peak current and the concentration of aspartame, respectively. The limit of detection (LOD) is calculated as per the following equation using three times noise, viz., signal/noise=3 (S/N=3) [54]:

LOD=KS/r                                (1)

where S is the standard deviation of signals measured 20 times for blank solution, and it is about 3.682×10-10 A for C1, 4.320×10-11 A for C2 and 1.547× 10-9 A for C3; K is a coefficient based on a certain confidence level factor (K=3); and r is the sensitivity of the detection method, viz., the slope of the calibration curve (Here, r=0.0354, 0.0960 and 0.0483 A M-1 for C1, C2 and C3, respectively). The LOD values are estimated to approximately be 3.12×10-8, 1.35×10-9 and 9.61×10-8 mol L-1 corresponding to the C1, C2 and C3 modified electrode sensor, respectively. ”.

To convince the readers that the proposed approach is advantageous, aspartame detection with nonmodified electrode/modified only by PMAEFc (no MWCNTs-ZnOs NPs) should be performed.

► Your comment is very valuable for improving the quality of our manuscript. Based on your comment, we have performed aspartame detection with nonmodified electrodes and modified only by PMAEFc (no MWCNTs-ZnOs NPs) to convince the readers the advantage of the proposed approach.

 In fact, during the CV measurement, the bare GCE (nonmodified electrodes), as well as the modified GCEs with MWCNTs, MWCNTs@ZnO and MWCNTs/PMAEFc, fails to generate any obvious redox response in PBS solution of pH=7, as shown in Figure 5(A)(a-d) in our manuscript. Therefore, C1, C2 and C3 modified electrodes are selected for the aspartame detection, and the nonmodified electrodes for the detection are meaningless. Despite this, the aspartame detection is conducted with nonmodified electrodes and the PMAEFc modified electrodes, and the results are shown in Figure 7(D). It is clear that the aspartame detection with both the nonmodified electrodes and the PMAEFc modified electrodes offers poor DPV curves, which produces irregular nonlinear calibration equations, as expected. Therefore, the detection based on the nonmodified and PMAEFc modified electrodes is low-reliability and meaningless compared with their C1, C2 and C3 counterparts, whereas the proposed approach is advantageous for the aspartame detection.

 Pages 10 and 11, lines 403-406: "... Consequently, it is concluded that the synergistic effect among the components may be responsible for good electrochemical response performances, and the C1, C2 and C3 modified electrodes are reasonably selected for the following analysis including aspartame detection.".

 Page 13, lines 492-458: "... In contrast, under the same conditions, the aspartame detection with both the nonmodified electrodes and the PMAEFc modified electrodes offers poor DPV curves, with no peak current, as shown in Figure 7(D), which produces irregular nonlinear or nonproportional calibration equations, as expected (not shown). Therefore, the detection based on the nonmodified and PMAEFc modified electrodes is low-reliability and meaningless compared with their C1, C2 and C3 counterparts, whereas the proposed approach is advantageous for the aspartame detection.".

The detection of the aspartame in real samples should be expanded in the manuscript, as well. Furthermore, practically no discussion is present; the authors should discuss the advantages and disadvantages of their approach with other methods presented in the literature.

► Thank you for your kind comment, and we have expanded the detection of the aspartame in real samples such as mango juice, kiwi fruit juice, strawberry juice and Lipton lipya tea sachet. Based on these detections, we have discussed the advantages and disadvantages of this approach with other methods in the literature. The related amendment can be found in our revised manuscript. See also Table 1.

Page 6, lines 235-249: “The two carbonate beverages (Coke zero and Sprite Zero), three fruit juices (Mango juice, Kiwi fruit juice and Strawberry juice) and Lipton lipya tea sachet were selected as real sample for residual aspartame detection. For carbonate beverages, their aqueous samples were ultrasonicated for 30 min and diluted with deionized water. Subsequently, the Coke Zero and Sprite Zero were directly spiked with appropriate amounts of aspartame standard solution. Specifically, a 50 ml 1×10-3 mol L-1 aspartame in PBS (pH=2) solution was prepared, and then the solution of 5 ml was mixed with 100 ml Coke and 100 ml Sprite zero to get a 14 mg l-1 (4.76×10−5 mol L-1) aspartame solution. As for fruit juice, 0.3 mg of each sample was weighed and diluted using 20 ml PBS (pH=2), whilst Lipton lipya tea sachet (2 g) was fetched out and dissolved in deionized water, and then transferred in a 1 L volumetic flask. For reducing the error, all solutions were filtered via a 0.45 µm membrane before measurements. The aspartame concentration was determined by the electrochemical modified electrodes with the method of DPV. The percentage of recovery was determined for 5 different concentrations, and relative standard deviations (RSDs) were measured for 20 times detection for the same concentration.”.

Page 14, lines 509-423: “For this purpose, the Coke zero, Sprite zero, mango juice, kiwi fruit juice, strawberry juice and Lipton lipya tea sachet sample solutions of 50 ml, which contain aspartame, were stirred for 30 min and served as stock solutions with no dilution. Then, the solution was added to the sample cells and the alteration in current density was detected by DPV. The electrochemical signals of each sample were recorded five times and averaged, with the RSD as low as 0.67, 0.17, 2.6, 1.8, 3.3,  and 4.1% for Coke zero, Sprite zero, mango juice, kiwi fruit juice, strawberry juice and Lipton lipya tea sachet samples, respectively. The concentration of aspartame in Coke zero, Sprite zero, mango juice, kiwi fruit juice, strawberry juice and Lipton lipya tea sachet was calculated using the calibration graph to be approximately 35.36 µM, 40.20 µM, 0.96 mM, 1.34 mM, 0.88 mM and 1.53 mM, respectively in this way. Compared with any other method presented in the literatures (Table 1) [28,56-64], this approach offers significantly low RSD and can expand the detection of the aspartame in real samples. Consequently, the developed MWCNTs@ZnO/PMAEFc/GCE modified electrode sensors are proved to be an effective tool for quick and simple detection of aspartame contents in real beverage or food samples.”.

Round 2

Reviewer 1 Report

The authors have really improved the clarity of the manuscript and I am almost convinced with the work. I am just not fully convinced with the answer on use of MWCNTs/PMAEFcas a control.

The authors mention that MWCNTs/PMAEFc failed to produce any redox peaks but did not give any reasoning to that? Could the authors describe why do they not see any redox peaks or are the redox peaks too small to measure?

Since Authors are using a new material, maybe authors can also look into other characterisation tools like XPS or AFM to look into both composition and layer thickness and morphology. Maybe that could address the point as to why peaks are not observed with just MWCNTs. The other reason could also be the alignment of MWCNTs as there are papers that have reported that MWCNTs needs to be aligned in a certain way to be conductive because if not aligned properly, it will be insulating.

There are quite a lot of papers on the use of ferrocene based compounds with CNTs on electrodes and have shown excellent ferrocene peaks. It is a little surprising that no peaks are observed. 

Author Response

Article reference: nanomaterials-591287

Article title: Organic-inorganic nanohybrid electrochemical sensors from multi-walled carbon nanotubes decorated with zinc oxide nanoparticles and in-situ wrapped with poly(2-methacryloyloxyethyl ferrocenecarboxylate) for detection of the content of food additives

Authors: Jing-Wen Xu, Zhuo-Miao Cui, Zhan-Qing Liu, Feng Xu, Ya-Shao Chen, Yan-Ling Luo

Response to Referee #1

The authors mention that MWCNTs/PMAEFc failed to produce any redox peaks but did not give any reasoning to that? Could the authors describe why do they not see any redox peaks or are the redox peaks too small to measure?

Since Authors are using a new material, maybe authors can also look into other characterisation tools like XPS or AFM to look into both composition and layer thickness and morphology. Maybe that could address the point as to why peaks are not observed with just MWCNTs. The other reason could also be the alignment of MWCNTs as there are papers that have reported that MWCNTs needs to be aligned in a certain way to be conductive because if not aligned properly, it will be insulating.

There are quite a lot of papers on the use of ferrocene based compounds with CNTs on electrodes and have shown excellent ferrocene peaks. It is a little surprising that no peaks are observed. 

► Thank you for your valuable comment, which is very important for improving our manuscript. In line with your comment, we have supplemented XPS and AFM experiments and explained the experimental phenomenon, and made corresponding revisions or corrections on our manuscript. We wish the revised paper will be satisfactory and finally be accepted by Nanomaterials. Thank you again!

Page 5, lines 197-202: “X-ray photoelectron spectroscopy (XPS) spectra were recorded on an AXIS ULTRA spectrometer (Kratos Analytical Ltd., Shimadzu Corp., Japan) at a voltage of 15 kV and room temperature.

Atomic force microscope (AFM, Dimension ICON, Brucker Corp., Germany) was employed to observe surface topography and distribution of MWCNTs@PMAEFc at a scan rate of 0.977 Hz, and to estimate the layer thickness of MWCNTs@PMAEFc.”.

Pages 10 and 11, lines 371-399: “However, the modified electrodes based on only any single material, even the simple combination of two components including MWCNTs, PMAEFc and ZnO NPs surprisingly do not show well-defined redox peak currents, which is different from the results reported by quite a lot of references in that ferrocene based compounds with CNTs modified electrodes have shown excellent ferrocene peaks [50-52]. It has been reported that the electrical properties of the composites depend on the aspect ratio, alignment and even alignment thickness, and dispersion of conductive fillers, and the alignment of MWCNTs in a certain way, viz. the alignment direction, is conductive; or else, the MWCNTs will be insulating [53,54]. Therefore, the reason why no redox peak is observed for MWCNTs and MWCNTs@PMAEFc is probably due to the random and nondirrectional alignment of MWCNTs without the field, as demonstrated in Figures 3 and 4. Moreover, the one-dimensional disorderly aligned MWCNTs have a low layer thickness. These factors result in poor conducting properties of MWCNTs and ferrocene-containing MWCNTs@PMAEFc nanocomposites.

XPS and AFM were further used to interpret the electrochemical response, as illustrated in Figure 5(B) and 5(C). XPS survey spectra in Figure 5(B) show the surface elemental compositions of the sample to be determined. MWCNTs produce two characteristic photoelectronic signals at binding energies (BEs) of 285.3 and 533.1 eV assigned to the C 1s and O 1s signals, respectively. The appearance of Fe 2p at 708.5–721.8 eV and Fe 3p peaks at 53.4 eV in Figure 5(B)(b) indicates that the ferrocene-containing polymer moieties were covered on the surface of the MWCNTs, forming MWCNTs/PMAEFc, with the Fe mass content of about 8.6%. The AFM image in Figure 5(C) clearly reveals that the MWCNTs@PMAEFc forms a randomly-aligned, chaotic and aggregated morphology with a mesh-like structure, and a total layer thickness is about 48-69 nm. The thickness of PMAEFc layers is estimated to be ca. 18-39 nm based on the size supplied by SEM and TEM as well as producer; these results are in accordance with those by TEM and SEM. The MWCNTs@PMAEFc nanocomposites with this kind of the randomly-arranged morphology, as stated above, would lead to poor conductivity and thus poor electrochemical redox response [53,54]. On the other hand, in comparison with the bulk PMAEFc moieties, the relatively low PMAEFc composition based on XPS result and layer thickness in MWCNTs@PMAEFc is maybe also   responsible for the poor current response.”.
